# IncomeSCM: From tabular data set to time-series simulator and causal estimation benchmark

**Fredrik D. Johansson**
Chalmers University of Technology
and University of Gothenburg
`fredrik.johansson@chalmers.se`

## Abstract

Evaluating observational estimators of causal effects demands information that is rarely available: unconfounded interventions and outcomes from the population of interest, created either by randomization or adjustment. As a result, it is customary to fall back on simulators when creating benchmark tasks. Simulators offer great control but are often too simplistic to make challenging tasks, either because they are hand-designed and lack the nuances of real-world data, or because they are fit to observational data without structural constraints. In this work, we propose a general, repeatable strategy for turning observational data into sequential structural causal models and challenging estimation tasks by following two simple principles: 1) fitting real-world data where possible, and 2) creating complexity by composing simple, hand-designed mechanisms. We implement these ideas in a highly configurable software package and apply it to the well-known Adult income data set to construct the `IncomeSCM` simulator. From this, we devise multiple estimation tasks and sample data sets to compare established estimators of causal effects. The tasks present a suitable challenge, with effect estimates varying greatly in quality between methods, despite similar performance in the modeling of factual outcomes, highlighting the need for dedicated causal estimators and model selection criteria.

## 1  Introduction

Estimating causal effects of interventions is fundamental to improving decision-making by data-driven means. As a result, an ever-growing body of research develops machine learning algorithms for this task (Hernán and Robins, 2006; Shalit et al., 2017; Wager and Athey, 2018; Yoon et al., 2018; Künzel et al., 2019; Kennedy, 2023). However, identification of causal effects—and, therefore, evaluation of these algorithms—rely either on interventions (e.g., randomized experiments) or unverifiable assumptions (e.g., exchangeability) (Pearl, 2009). As experimentation is costly or prohibited in many domains, simulated environments are uniquely suited as benchmarks for causal effects estimation from observational data. They can be constructed to satisfy conditions for identifiability and to answer interventional queries, providing ground truth labels for estimates. But this strategy begs the question: How can we build simulators that reveal which causal estimators work best in the real world?

Primarily, two kinds of data sets are used for offline evaluation of causal effect estimates: experimental data and (semi-) synthetic data. Experimental data (e.g., the LaLonde (Jobs) data set (LaLonde, 1986)) supports unbiased estimation by removing confounding by construction (Imbens and Rubin, 2015). But learning from unconfounded data is rarely the goal of the methods we seek to evaluate. To overcome this, researchers can introduce selection bias synthetically by subsampling observations depending on the intervention, as in Twins (Louizos et al., 2017). A drawback of this approach is that experimental data tends to be small, both in sample size and dimensionality, and subsampling decreases its size even further. In other semi-synthetic data, e.g., IHDP (Hill, 2011), the analyst simulates one or more of the variables in the system of interest, typically the outcome variable.

Hand-designed simulators are great for designing benchmarks to test estimators in a particular challenge, such as learning from non-linear outcomes (Hill, 2011) or heteroskedastic noise (Hahn et al., 2019). Their main drawbacks are that they (i) scale poorly with the size of the system, requiring significant domain expertise by the designer, or (ii) model overly simple or extreme systems, not reflecting the intricacies of real-world data (Hernán, 2019). In contrast, purely data-driven simulators of potential outcomes can fit and generate complex relations between context, treatment, and outcome variables which are difficult to design by hand (Chan et al., 2021). However, they often lack structural constraints on the causality between variables, making them insufficient for reasoning about rivaling identification strategies, e.g., with instrumental variables or alternative adjustment sets, and for changing the nature of confounding or interventions (Hernán et al., 2019). In short, mimicking only the observational distribution of a chosen system is not sufficient to stress test causal effect estimators.

**Contributions.** In this work, we construct a simulator using a sequential structural causal model based on the well-known Adult data set (Becker and Kohavi, 1996) and use this to create a new benchmark data set for causal effect estimation. We apply a generalizable strategy for developing realistic and challenging estimation tasks by imposing structural constraints on the simulator using a causal graph and fitting parts of its mechanisms to non-sequential tabular data. Our strategy assumes that the initial state of a system of variables follows a complex distribution (e.g., the age, education, and income of a subject), but that its transitions are simpler (e.g., the income next year is close to the income from the previous year). Thus, we can build a realistic simulator by fitting the causal mechanisms of initial states to data and hand-designing transition mechanisms. The effects of interventions in early time steps on outcomes later in time are determined by compositions of several transitions which renders causal effects non-trivial functions of the initial state, even if each transition is simple. We use our simulator `IncomeSCM` to generate data for an observational study of the effects of `studies` on future `income` and compare popular estimators from the literature. Our results indicate that different estimators yield estimates that differ substantially in quality despite fitting observed variables similarly well, pointing to the challenging nature of the tasks.

## 2  Related work

Simulated data has long been used to evaluate estimators of causal effects as a way to get around the unavailability of counterfactual outcomes. The extent and nature of simulation vary from replacing a single variable (typically the outcome) with a simulated counterpart, to simulating an entire system of random variables. A notable example, the "IHDP" data set was derived from a randomized study of the Infant Health and Development Program by removing a biased subset of treated participants and introducing a synthetic response surface (Hill, 2011). It was built to support multiple settings, see e.g., (Dorie, 2016), but the variant used by Shalit et al. (2017) has stuck as the primary benchmark. Generally, randomized data can be turned observational by subsampling cohorts with treatment selection bias, as in Twins (Louizos et al., 2017) or by adding observational data to an experimental cohort as in Jobs (Shalit et al., 2017). Both strategies benefit from context variables remaining untouched, representative of real-world data. A downside is that the sample size is fixed to the number in the original study. This limitation can be overcome by designing every variable in the system, such as in the 2016 data challenges of the Atlantic Causal Inference Conference (ACIC) (Dorie et al., 2019). However, this often leads to very simplistic simulators.

Recognizing the limitations of relying on purely synthetic simulators for benchmarking, several works have proposed fully or partially data-driven simulators, where components are learned from real-world data. A recent example is Medkit-Learn Chan et al. (2021), developed for off-policy evaluation of reinforcement learning, and the Alzheimer's Disease Causal estimation Benchmark (ADCB) (Kinyanjui and Johansson, 2022). The 2022 iteration of the ACIC challenge included a partially data-driven design, which used a strategy similar to ours, but to the best of our knowledge, the details of the simulator remain unpublished (Mathematica, 2022). Complementing purpose-built data sets and simulators, causality toolboxes, such as DoWhy (Sharma and Kiciman, 2020) and CausalML (Chen et al., 2020) include multiple ways of generating simple forms of synthetic data. Gentzel et al. (2019) provide an excellent overview of different approaches to evaluating causal modeling methods and suggest another approach: Use complex systems, such as a PostGres database, where interventions on its configuration can be performed but the outcome are hard to predict. This can provide both observational data for learning and interventional data for evaluation. Appendix D contains a tabular summary of common causal effect estimation benchmarks.

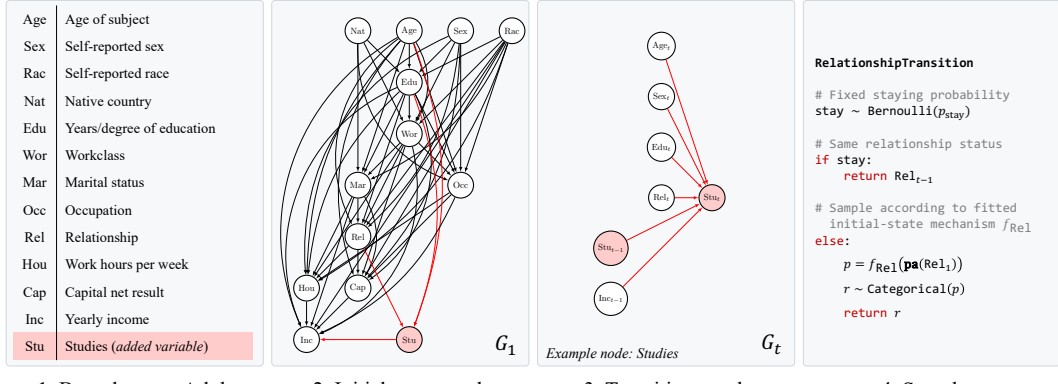

| Age | Age of subject |
| Sex | Self-reported sex |
| Rac | Self-reported race |
| Nat | Native country |
| Edu | Years/degree of education |
| Wor | Workclass |
| Mar | Marital status |
| Occ | Occupation |
| Rel | Relationship |
| Hou | Work hours per week |
| Cap | Capital net result |
| Inc | Yearly income |
| Stu | Studies (*added variable*) |

```
RelationshipTransition

# Fixed staying probability
stay ~ Bernoulli(p_stay)

# Same relationship status
if stay:
    return Rel_{t−1}

# Sample according to fitted
  initial-state mechanism f_Rel
else:
    p = f_Rel(pa(Rel_1))
    r ~ Categorical(p)
    return r
```

1. Base data set: Adult    2. Initial-state graph    3. Transition graph    4. Samplers

Figure 1: Overview of the IncomeSCM simulator. All variables (left) are observed at a single time point in the well-known Adult data set, except the Studies variable (node and edges in red). A causal graph is constructed for the initial state (middle-left) and for the transitions, illustrated here by the parents of a single node in the full graph (middle-right). Each variable is associated with an initial sampler and a transition sampler (example to the right) which simulate the trajectory of the variable.

Finally, several previous works have analyzed the Adult data set in the context of causality (Von Kügelgen et al., 2022; Mahajan et al., 2019), for example, in causal discovery (Binkytė et al., 2023). As a result, there are several DAGs proposed in the literature, including those by Zhang et al. (2016) (inferred from data), by Li et al. (2016) and by Nabi and Shpitser (2018) in the context of counterfactual (algorithmic) fairness (constructed by the analysts). We use the latter strategy here.

## 3  Constructing the simulator

We create `IncomeSCM-1.0`—a simulator of a sequential decision-making process associated with personal income. This will serve as an example of a general strategy for creating realistic and challenging benchmark tasks for causal effect estimation from cross-sectional observational data.

`IncomeSCM` is based on a discrete-time Markov structural causal model (SCM). The primary goal of the model is to simulate the effects of an intervention of interest $A$ on a given outcome $Y$ in a system of random variables $\mathcal{V} := \{A, Y\} \cup \mathbf{X}$, where $\mathbf{X}$ is a set of context variables. Each variable $V \in \mathcal{V}$ is simulated at $T$ time points in a sequence $V_1, ..., V_T$. The *initial state* of $V_1$ is determined by a deterministic structural equation $V_1 := f_V(\mathbf{pa}(V_1), U_{V_1})$ where $\mathbf{pa}(V_1)$ are the *parents* (direct causes) of $V_1$ and $U_{V_1}$ is a source of noise. Values of $V_t$ for $t > 1$ are determined by a time-homogenous *transition* mechanism $V_t := g_V(\mathbf{pa}(V_t), U_{V_t})$. In general, mechanisms and parent sets are *not shared* by initial states and transitions, i.e., $f_V \neq g_V$ and $\mathbf{pa}(V_1) \neq \mathbf{pa}(V_t) = \mathbf{pa}(V_s)$ for $s, t > 1$. One of the main reasons for this is that transitions $g_V$ typically depend on the previous value of the variable $V$, but the initial state cannot. As a result, the initial state of a variable typically has a much stronger dependence on other variables in the systems than the transitions do. The model is Markov in that $\mathbf{pa}(V_t)$ includes only variables from time $t$ and $t - 1$.

To ensure that the model captures realistic correlations between variables, its structural equations are *learned* whenever data on a variable and its parents are available. When such data are unavailable, the structural equations are *hand-designed*. Our simulator and data set construction, generalizable to any cross-sectional data set, proceeds as follows (see Figure 1 for an overview):

1. **Data set.** Compile a base cross-sectional data set $\mathcal{D}$ with observations of random variables $\mathcal{V}$ from which to learn the structural equations of the simulator.

2. **Initial-state graph.** Learn or design, based on domain knowledge, a causal directed acyclic graph (cDAG) $G_1$ of the initial states $V_1$ of all variables $V \in \mathcal{V}$.

3. **Transition graph.** Posit a time-independent cDAG $G_t$ for the causes of transitions of variable values as functions of variables in the current and previous time steps.

4. **Samplers.** Learn samplers (structural equations) $f_V$ of the initial states of each variable $V$ by fitting estimators to the base data set and design samplers $g_V$ for transitions.

5. **Simulation.** Create observational and counterfactual data sets of causal effects from the SCM through ancestral sampling, while performing interventions on one or more variables.

A benefit of this strategy is that it blends the strengths of data-driven and knowledge-guided design: 1) Imposing a causal structure defines interventional distributions of all variables, allowing benchmarks to ask flexible causal queries of the simulator (e.g., about mediation or path-specific effects), 2) Fitting the initial states of variables adds realistic complexity to structural equations. 3) A sequential model creates causal effects of future outcomes that are interesting functions of early interventions and context variables even when transition mechanisms are hand-designed.

**On hand-crafted transitions.**    The design strategy for our simulator is developed for systems with simpler transitions than initial states. A strong argument for this idea is that most variables are easier to model once you know a recent value of them. The classical example is that the weather tomorrow is usually similar to the weather today (transition is simple), but modeling the weather based on other variables, such as location and time of year, is more difficult (the initial-state model is complex). Another example is the distribution of patients in a hospital. When a new patient gets admitted (initial state), we can view their demographics, symptoms, and medications as drawn from a complex distribution of possible patients. When we observe them the next day, the changes in their variable tend to be more predictable given their values at admission and obey simpler rules, such as dose response to medication or day-night cycles of vitals. A counter-example to this pattern would be a system where all variables are independent in the initial state but become dependent in transitions. For example, let the variables measure the opinions of a random set of people (e.g., students) who begin to meet regularly (say, in a class). In the initial state, there will be no correlation among the opinions but he next state may be a complicated function of the opinions of different people.

As an example of a generic task derived from such a simulator, we consider atomic interventions $A_{t_0} \leftarrow a$ performed at a baseline time point, $t = t_0$, on an outcome $Y_T$, measured at the final time step, $t = T$. In the potential outcomes framework (Imbens and Rubin, 2010), we define the conditional average treatment effect with respect to a subset of baseline variables $Z \subseteq X_{t_0}$,

$$\text{CATE}(z) = \mathbb{E}[Y_T(A_{t_0} \leftarrow 1) - Y_T(A_{t_0} \leftarrow 0) \mid Z = z] \,.$$

and aim to estimate CATE from observational data generated by the simulator, as well as the average treatment effect, $\text{ATE} := \mathbb{E}_Z[\text{CATE}(Z)]$.

### 3.1   The base data set: Adult

`IncomeSCM-1.0` is based on the widely-used Adult data set (Becker and Kohavi, 1996) and its release on the UC Irvine Machine Learning Repository.[1]  The set contains 48 842 observations of 14 variables and a sample weight, extracted from the 1994 US Census Bureau database. The original task associated with the Adult set is to predict whether an individual will earn more than $50 000 in a year and the full variable set comprises `age`, `workclass`, `education` (categorical and numeric), `marital-status`, `occupation`, `relationship`, `race`, `sex`, `capital-gains/losses` (summarized as `capital-net` in the simulator), `hours-per-week` (of work), `native-country` and `income`. We ignore the `sample-weight` in the simulator and drop any rows containing missing values in any column, leaving 30 162 samples. Statistics for these features are presented in Table 1.

**Outcome of interest**    As the primary outcome of interest $Y$, we use the yearly income of subjects in US dollars. In the Adult data set, income is coded as a binary variable, representing whether the yearly income of a subject exceeds $50 000, denoted here by $R \in \{0, 1\}$. For `IncomeSCM`, we construct a continuous `income` variable $Y$ by first fitting a small random forest *regressor* $h_R$ to predict the original binary outcome $R$ as a function of all covariates $X$ (excluding the added variable `studies`). The continuous predictions $\hat{R}_i = h_R(X_i) \in [0, 1]$ of the regressor are computed for each subject $i$. This intermediate value represents the likelihood that the income exceeds $50 000, which is then re-scaled and shifted by two constants so that the average of the resulting numbers, across the full cohort, matches the average US population annual salary (in USD). Leaving the outcome a binary

---

[1]`https://archive.ics.uci.edu/dataset/2/adult`

variable would make it more difficult to design, for example, income transitions. With a continuous value, we can reason about the average increase in salary due to a pay raise. A binary-income model would be much more coarse-grained.

**Intervention of interest**   For a causal estimation task to be realistic, the primary intervention of interest must be possible to implement in practice, for example, in an experiment (Hernán, 2019; Hernán et al., 2022). The Adult data set contains a small number of variables that could be *directly* intervened upon and serve as the primary intervention. Base variables `age`, `sex`, `native-country`, and `race` can be removed from consideration, as can `capital-gain`, `capital-loss` as these are outside of the control of any feasible experiment. `occupation` is feasible to change on a yearly basis, but most jobs are unavailable to most people. `relationship` and `marital-status` are possible to change, through e.g., divorce, but most people would not be willing to change these in order to optimize their `income`, let alone participate in an experiment to do so. `workclass`, `occupation` and `hours-per-week` are candidates that could be intervened upon a yearly basis. `education` is coded in Adult by the highest attained degree (e.g., Bachelor's degree) or the number of years of study. However, in the time scale of the simulator (one observation per year, $t = 1, ..., T$), an intervention of, say $A_t \leftarrow \text{Doctorate}$ is not feasible. Instead, to showcase the blend of data-driven and knowledge-guided design, we create an intervention on `studies`, a constructed variable, with values "Full-time studies", "Day course", "Evening course" and "No studies", representing the study activity of a person during a year. The initial state of `studies` is determined by a logistic regression with manually selected coefficients for `age`, `relationship`, and `education`, and intercepts for each of the study types. `studies` directly affect the education level, current `income` and the propensity to study in the next year, and indirectly the future income of a subject.

## 3.2   The causal graph

We construct a cDAG for the Adult data set (*the* initial-state cDAG for `IncomeSCM-1.0`) by assigning direct causes to each variable. `age`, `sex`, `race`, and `native-country` were taken to be base variables without causal parents in the data set. `education` was deemed to be causal of `workclass`, which in turn causes `occupation` and `marital-status`. `marital-status` and `relationship` are closely linked, and the causal direction was chosen to be from the former to the latter. Finally, `hours-per-week` and `capital-net` were determined by the previous variables, and `income` is the furthest downstream, directly caused also by `studies`, the hand-designed variable. The full graph is illustrated in Figure 1 (middle-left). A very similar graph was used by Nabi and Shpitser (2018) except, in their case, `marital-status` was a direct cause of `education`, not the other way around.

For transitions, the full causal graph is too large to be easily legible. Instead, we give an example in Figure 1 (middle-right) and report all direct causes as an edge list in Table 6 in Appendix E. Each variable has transition causes that come either from the previous time step, e.g., $\text{age}_{t-1} \rightarrow \text{age}_t$, or from the current time, e.g., $\text{studies}_t \rightarrow \text{income}_t$. Base variables `age, sex, race` and `native-country` depend only on their previous values.

## 3.3   The samplers

IncomeSCM implements structural equations through *samplers*, functions of the values of parent variables, and independent sources of noise. The variables are either numeric or categorical. For the initial states of numeric variables $V \in \mathbb{R}$, we use (with few exceptions) an additive noise model,

$$V = f_V(\mathbf{pa}(V)) + \epsilon_V ,$$

where $f_V$ predicts the value of $V$ given its parents—$f_V$ is a model of the conditional mean, $\mathbb{E}[V \mid \mathbf{pa}(V)]$—and $\epsilon_V$ is a source of homoskedastic noise (e.g., Gaussian). All structural equations for initial-state variables, except for `studies`, have observations $(V, \mathbf{pa}(V))$ available (`income` is a special case as described above). For these, $f_V$ is implemented by fitting a regression (e.g., a Random forest regressor) to the observed data, and the scale of $\epsilon_V$ is chosen proportional to the mean squared regression error. `capital-net` is an exception to this form, since the majority of capital gains/losses is 0. For this, we blend a classifier that predicts whether the value will be non-zero and a conditional regressor for the case where the value is non-zero.

For categorical variables $V \in \{1, ..., k\}$, we sample from a categorical distribution given by the softmax function $\sigma$ applied to map $f_V$ from parent variables to logits for the categories of $V$. By the

Gumbel-Max trick (Gumbel, 1954), we can write this as a structural equation

$$V = \arg\max_j f_V(\mathbf{pa}(V))_j + \epsilon_j \,,$$

where $\epsilon_j \sim \mathrm{Gumbel}(0, 1)$. For observed pairs $(V, \mathbf{pa}(V))$, the map $f_V$ is fitted using stochastic classifiers, such as logistic regression, classification trees or neural networks with softmax outputs.

**Hand-crafted samplers.** For equations with no available data, such as for `studies` and all variable transitions, the samplers are designed by hand. All transition samplers are described in Appendix Table 7. For the intervention variable `studies`, the initial-state distribution and transitions are described above. The demographic variables `native-country`$_t$, `sex`$_t$ and `race`$_t$ are constant, and `age`$_t$ increments deterministically by 1 each time step. For the initial state of the outcome variable `income`$_1$ and the transitions $g_V$ of most variables $V \in \mathcal{V}$, we use a blend of hand-designed rules and fitted regressions. This allows us to generate complex dependencies between variables without observing a time series. For example, `income`$_1$ is based on a regression of the original binary variable in the Adult data set, with hand-crafted modifications depending on `studies` and `workclass`. The transition sampler of `relationship`$_t$ has a fixed probability $p_{\mathrm{stay}}$, that the person will stay in their relationship, `relationship`$_t$ = `relationship`$_{t-1}$. With probability $1 - p_{\mathrm{stay}}$, `relationship`$_t$ is resampled from the same fitted model as the initial state, see Figure 1 (right). In this way, only the event that the relationship is changed is hand-designed, but what it is changed to depends on data. A similar strategy is used for all context variables except demographics and `education`.

## 4 Implementation & data set

`IncomeSCM-1.0` is implemented in Python as a "fit-predict" estimator in the style of scikit-learn (Pedregosa et al., 2011). The code for the simulator and for reproducing all experimental results is available at `https://github.com/Healthy-AI/IncomeSCM`. The simulator is configured by a YAML specification of all variables in the system, how they should be learned from data, and how they evolve in a time series. The specification is passed to the simulator at initialization and is used to fit structural equations to observed data from the base data set. We use the variable `education` as an example of such a variable specification below.

```
education:
  parents:  [age, race, sex, native-country]
  sampler:
    type:  LogisticSampler
    multi_class:  multinomial
  seq_parents_curr:  []
  seq_parents_prev:  [education, studies]
  seq_sampler:
    type:  EducationTransition
```

Here, `parents` specify the direct causes of the initial state of the variable. `sampler` specifies the type and parameters of the sampler. In this case, the initial state of `education` is distributed according to a logistic regression of `age, race, sex` and `native-country`. `seq_parents_curr` specifies the direct causes of the transition probability from the current time step, and `seq_parents_prev` causes from the previous time step. `seq_sampler` specifies which function determines this probability. `LogisticSampler` is a general sampler, that outputs the probability of the given variable as a logistic-linear function of the direct causes specified in `parents`. General samplers can be reused for other data sets and are defined in `samplers.py`.

Since the base data set is cross-sectional (from a single time point), `EducationTransition` is a hand-crafted sampler that returns the distribution of the education variable in the next time step. In this example, `Education` is determined completely by `studies`, depending on the type of studies and with a small probability of not advancing due to failing the year.

### 4.1 Benchmark data set

We construct a single-time causal effect estimation benchmark around the question:

> What is the effect on the `income` $Y_7$ of a subject at $t = 7$, six years after a year of $A_2 =$"Full-time studies" compared to $A_2 =$'No studies" at $t = 2$, and then proceeding as they wish?

For this task, we create a cross-sectional observational data set by first sampling from the simulator at time $t = 1, ..., 7$ where `studies` are selected by a "default" observational stochastic policy as defined by the samplers in IncomeSCM. Then, time $t = 1$ is used to record the previous income level `income_prev` and studies `studies_prev`, prior to intervention. All other covariates are read off at time $t_0 = 2$. The binary intervention of interest is then performed at $t_0 = 2$, with $A_2 = 1$ representing `Full-time studies` and $A_2 = 0$ `No studies`. The outcome variable $Y_T =$`income`$_T$ is recorded at $T = 7$. We use this to sample $n = 50000$ independent and identically distributed observation sequences, each representing an individual. The sequences are drawn with an incrementing random seed, starting at $s = 0$. First-order covariate statistics of covariates, interventions, and outcomes for this seed can be seen in Table 1. There are no unobserved confounders or temporal confounding in this benchmark task—all direct causes of the intervention are observed in the cross-sectional data set.

To create ground truth for causal effect estimates, we generate two analogous sets of trajectories where the same set of subjects $i = 1, ..., m$ are either *all* assigned $A_2^i = 1$ (`Full-time studies`) or *all* assigned $A_2^i = 0$ (`No studies`), thereby recording samples of *both* counterfactual outcomes $Y^i(0), Y^i(1)$ of these subjects. We control that the sets of subjects are identical before both interventions at $t_0 = 2$ by setting a shared seed, $s = 1$. We use a different seed from the observational data set to measure out-of-sample generalization. The resulting data set(s) of observational and two interventional samples is referred to as `IncomeSCM-1.0.CATE`.

Identification strategies for the causal effect of any variable in the simulator on any downstream variable can be derived from the edge list in Table 6 using graphical criteria like the backdoor criterion (BDC) (Pearl, 2009). In the benchmark data set, the observed intervention is performed at time $t = 2$ with $t = 1$ being the initial state. By the BDC, a minimal adjustment set for estimating the causal effect of intervening on `studies`$_2$ is the set of direct causes of this variable. In `IncomeSCM-1.0`, `studies`$_t$ at times $t > 0$ are directly caused by `studies`$_{t-1}$, `income`$_{t-1}$, `age`$_t$, `sex`$_t$, `education`$_t$, and `relationship`$_t$.[2] Regression adjustment of potential outcomes or propensity-based adjustment using these variables leads to the identification of causal effects. Expanding this set with any pre-intervention variables (e.g., other variables at times $t = 1, 2$) also results in valid adjustment sets.

We consider three tasks on the same dataset based on the following causal estimands:

Task 1. ATE and CATE conditioned on the full set of pre-intervention covariates, including previous income and previous studies.

Task 2. ATE and CATE conditioned only on the direct causes of the intervention $A_2$.

Task 3. CATE, conditioned only on `education`$_2$ (not itself a valid adjustment set), represented as 16 numeric bins, using the full adjustment set from Task 1.

Task 3 is an example of when the conditioning set of the CATE differs from any valid adjustment set. That is, to estimate the CATE conditioned only on `education`$_2$, it is not sufficient to adjust only for this variable; it requires a slightly unusual application of common causal effect estimators.

## 5  Experiments

The observational base statistics for `IncomeSCM-1.0.CATE`, as well as fitting results for samplers fit to data ($R^2$ for continuous variables, AUC for discrete variables) are presented in summarized form in Table 1. The full set of statistics for categorical variables is presented in the Appendix, Table 4.

We see a generally good match in the base statistics of variables, with the biggest discrepancy for `income>50k`. This is due to renormalizing the simulator samples to achieve an average income of \$70 000 before applying noise and correcting for the effects of `studies` and `workclass = Without-pay`. The discriminative performance for the fitted samplers (AUC/$R^2$) varies greatly between variables, some (such as `marital-status`) being more predictable from their parents in the posited causal graph than others (such as `capital-net`). This is expected since the original data

---

[2] In the implementation, the helper variable `time` is also passed to enable interventions at specific time steps.

Table 1: Baseline characteristics in the `IncomeSCM-1.0` cohort at $t = 0$ and the base data set Adult (after removing missing values), as well as in-sample fit quality. Categorical variables are reported as number (rate) and continuous variables as mean (interquartile range). ★The proportion of subjects with income higher than \$50 000 is higher in `IncomeSCM-1.0` than in Adult since the average income was calibrated to match the US population of 2023. †Base variables do not involve modeling, only computing proportions/means. *The `capital-net` regressor is fit only to sample with non-zero result. Fit quality is reported as $R^2$ for continuous variables and AUC for categorical variables. ‡The income variable is evaluated using AUC since the raw data has binary income indicators.

| Variable | `IncomeSCM-1.0` | Adult | $R^2$/AUC |
|---|---|---|---|
| $n$ | 50000 | 30162 | |
| native-country (+ 30 more categories) | | | † |
|    United-States | 45519 (91.0) | 27504 (91.2) | |
| sex = Female | 16206 (32.4) | 9782 (32.4 | † |
| race (+ 3 more categories) | | | † |
|    White | 42947 (85.9) | 25933 (86.0) | |
|    Black | 4677 (9.4) | 2817 (9.3) | |
| age | 41.1 (32.0, 49.0) | 38.4 (28.0, 47.0) | † |
| education (+ 14 more categories) | | | 0.68 |
|    HS-grad | 15086 (30.2) | 9840 (32.6) | |
|    Some-college | 9558 (19.1) | 6678 (22.1) | |
| workclass (+ 5 more categories) | | | 0.70 |
|    Some-college | 9558 (19.1) | 6678 (22.1) | |
|    Self-emp-not-inc | 3804 (7.6) | 2499 (8.3) | |
| occupation (+ 12 more categories) | | | 0.81 |
|    Prof-specialty | 6968 (13.9) | 4038 (13.4) | |
|    Craft-repair | 6626 (13.3) | 4030 (13.4) | |
| marital-status (+ 4 more categories) | | | 0.77 |
|    Married | 26695 (53.4) | 14456 (47.9) | |
| relationship (+ 4 more categories) | | | 0.92 |
|    Husband | 17263 (34.5) | 12463 (41.3) | |
|    Wife | 6767 (13.5) | 1406 (4.7) | |
| capital-net | 1081.1 (0.0, 0.0) | 1003.6 (0.0, 0.0) | 0.11* |
| hours-per-week | 42.0 (26.0, 57.0) | 40.9 (40.0, 45.0) | 0.28 |
| studies | | | |
|    Full-time studies | 6586 (13.2) | | |
|    No studies | 25241 (50.5) | | |
| income>50K (baseline) | 21167 (42.3)★ | 7508 (24.9) | |
| income_prev (\$1000) | 60.1 (16.1, 82.9) | | 0.93‡ |
| income (\$1000) | 78.2 (34.6, 103.1) | | |

set was not constructed with predicting *all* variables in mind, nor are all available variables used for prediction, only the causal parents. However, it does mean that the values of variables with low discriminative performance are largely determined by exogenous noise.

**Causal effect estimators.** We include regression estimators in the form of T-learners, which fit one model per potential outcome, and S-learners, which treat the intervention as an input to a single model, mixed in with adjustment variables (Künzel et al., 2019). For both, we compare multiple base estimators in ridge regression (Ridge), random forests (RF), and XGBoost (XGB). Second, we use two classical inverse-propensity weighting estimators, the Horvitz-Thompson (IPW) and Hayek or "Weighted" (IPW-W). Third, we include three double-machine learning (DML) estimators created by fitting regressions to the target $\frac{y_i - M_Y(x_i)}{a_i - M_A(x_i)}$, where $M_Y, M_A$ model $\mathbb{E}[Y \mid X]$ and $p(A = 1 \mid X)$ respectively, using sample weights $w_i = (a_i - M_A(x_i))^2$. This stems from the residualization of Robinson (1988), as used by Chernozhukov et al. (2018). The three variants use different base estimators of the nuisance functions. DML (Mix) uses an XGBoost regressor for $M_Y$, logistic regression for $M_A$, and linear regression for the cate estimator. Finally, we include a nearest-neighbor

Table 2: Results estimating CATE and ATE using the full `IncomeSCM-1.0.CATE` data set. $R^2$ is the coefficient of determination, AE is the absolute error. AUC/$R^2$ CV are the (non-random) 5-fold cross-validation AUC of the propensity estimate, and $R^2$ of the observed outcome, for propensity and regression estimators, respectively. For $R^2$ CATE and AE ATE, the confidence intervals ($\alpha = 0.05$) are computed by 1000 bootstrap iterations over test samples. An unbiased estimate of the ATE is \$40 919. Empty cells are intentionally left out as they don't apply to the corresponding estimators.

| Estimator | $R^2$ CATE (↑) | | AE ATE (\$ ↓) | | AUC/$R^2$ CV (↑) |
|---|---|---|---|---|---|
| "Full" adjustment set: All covariates | | | | | |
| IPW (LR) | | | 18645 | (18121, 19166) | 0.98 |
| IPW (RF) | | | 57025 | (56501, 57546) | 0.98 |
| IPW-W (LR) | | | 4878 | (4354, 5398) | 0.98 |
| IPW-W (RF) | | | 17406 | (16882, 17927) | 0.98 |
| Match (EU-NN) | | | 7147 | (6623, 7668) | -1.56 |
| S-learner (Ridge) | -0.00 | (-0.00, -0.00) | 400 | (22, 901) | 0.79 |
| S-learner (XGB) | 0.15 | (0.14, 0.15) | 533 | (90, 1023) | **0.84** |
| S-learner (RF) | 0.03 | (0.03, 0.03) | 487 | (50, 985) | **0.84** |
| DML (Linear) | -0.02 | (-0.02, -0.02) | 8299 | (7784, 8817) | 0.79 |
| DML (XGB) | -0.25 | (-0.26, -0.23) | 4580 | (4012, 5199) | 0.83 |
| DML (Mix) | 0.06 | (0.06, 0.07) | 6069 | (5575, 6581) | 0.83 |
| T-learner (Ridge) | 0.24 | (0.23, 0.25) | 1696 | (1213, 2164) | 0.80 |
| T-learner (XGB) | 0.24 | (0.23, 0.25) | **207** | (10, 574) | 0.83 |
| T-learner (RF) | **0.30** | (0.29, 0.31) | 2719 | (2256, 3165) | **0.84** |
| "Minimal" adjustment set: Direct causes of treatment only | | | | | |
| IPW (LR) | | | 12106 | (11582, 12627) | 0.97 |
| IPW (RF) | | | 58118 | (57594, 58638) | 0.98 |
| IPW-W (LR) | | | 3369 | (2845, 3890) | 0.97 |
| IPW-W (RF) | | | 16133 | (15609, 16653) | 0.98 |
| Match (EU-NN) | | | 8937 | (8413, 9458) | -1.56 |
| S-learner (Ridge) | -0.00 | (-0.00, -0.00) | 1701 | (1177, 2222) | 0.78 |
| S-learner (XGB) | 0.14 | (0.14, 0.14) | 1528 | (1020, 2012) | **0.80** |
| S-learner (RF) | 0.02 | (0.02, 0.03) | **1143** | (630, 1643) | **0.80** |
| DML (Linear) | **0.21** | (0.20, 0.22) | 2722 | (2239, 3177) | 0.78 |
| DML (XGB) | -1.13 | (-1.17, -1.10) | 16305 | (15558, 17033) | 0.79 |
| DML (Mix) | 0.18 | (0.17, 0.20) | 11887 | (11405, 12329) | **0.80** |
| T-learner (Ridge) | 0.20 | (0.18, 0.21) | 6104 | (5630, 6595) | 0.79 |
| T-learner (XGB) | -0.13 | (-0.15, -0.11) | 16549 | (16012, 17095) | 0.79 |
| T-learner (RF) | 0.16 | (0.14, 0.17) | 10931 | (10453, 11406) | **0.80** |

matching estimator with a Euclidean metric. The IPW, matching, and Ridge-regression S-learner estimators return estimates only of the ATE, not the CATE. For a more detailed description of all estimators, see Appendix F.

Hyperparameters for all estimators (see Appendix Table 8) were selected based on 5-fold cross-validation of 20 uniformly sampled hyperparameter settings. The estimators were then refitted to the entirety of the observational training data. Producing all results, including fitting the simulator, sampling, estimating, and evaluating causal effects, took less than 1 hour on a 2023 M2 MacBook Pro. The results were replicated on a 2 x 16 core Intel(R) Xeon(R) Gold 6226R CPU @ 2.90GHz with minimal differences except for XGBoost which is known to yield different fits for different machines.

**Results.** We present the results for causal effect estimation Tasks 1 & 2 in Table 2 and for Task 3 in Table 3 and Figure 2. IPW methods and S-learner (Ridge) are excluded from the latter two since these methods do not return heterogeneous effect estimates, only estimates of the ATE. The fit to observables was good for both propensity models (AUC $\approx 0.98$ in predicting observed treatments) and regression estimators ($R^2 \approx 0.80$ in predicting observed outcomes). Despite this, all causal estimation tasks were challenging for the methods to solve. The best methods achieved CATE $R^2$ of

Table 3: Results estimating CATE on `IncomeSCM-1.0.CATE`, conditioned only on `education` with the full adjustment set of Task 1. $R^2$ and 95% CI computed from 1000 bootstrap samples uses CATE estimated from counterfactual samples within each `education` bin as ground truth.

| Estimator | $R^2$ CATE$_{\text{EDU}}$ ($\uparrow$) |
|---|---|
| S-learner (XGB) | 0.33 (0.21, 0.44) |
| S-learner (RF) | -0.02 (-0.12, 0.07) |
| T-learner (Ridge) | 0.02 (-0.30, 0.27) |
| T-learner (XGB) | -0.36 (-0.75, -0.04) |
| T-learner (RF) | -0.61 (-1.25, -0.15) |

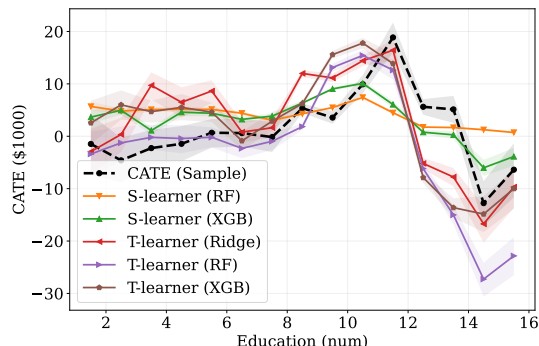

Figure 2: Estimated CATE conditioned on `education` (numeric) by stratifying CATE estimates w.r.t. the full conditioning set of Task 1.

only 0.30, 0.20, and 0.33 on Tasks 1, 2, and 3, respectively, falling short of explaining the majority of variation in these effects with the respective conditioning set. This is likely in large part to the separation between intervention and outcome by a horizon of six time steps (years), which introduces substantial uncertainty and functional complexity into the process.

No method is consistently at the top: The best CATE estimator is not the best ATE estimator and the best CATE estimators are different for Tasks 1, 2, and 3. More importantly, the best factual fit (e.g., S-learner (RF) in Task 1) does not yield the best effect estimates. Despite great discrimination of the fitted propensity scores, IPW estimators generally returned ATE estimates with large errors. These results show the importance of constructing several evaluation tasks to understand which methods are preferable in which circumstances. For example, in Figure 2, we see that S-learner (XGB) captures very little of the variation in CATE with `education`, despite passable performance in Task 1.

# 6 Discussion

We have presented the simulator and sequential structural causal model `IncomeSCM-1.0`, modeled on the Adult data set. The simulator exemplifies a general strategy for creating simulators with causal structure from cross-sectional data sets and use these to create benchmarks for causal effect estimation with challenging complexity. However, the current version and benchmark data set have several limitations. First, all trajectories are equally long, with a fixed horizon of $T$ years. This matches observational studies with a fixed follow-up window but is not representative of a common data analysis challenge: censoring. Specifically, no subjects leave the "study", either by choice or due to death. Since the age distribution of the initial state follows the census data behind the Adult data set, if the horizon $T$ is set to 100, the simulator would generate samples of humans well over 150 years of age. Similarly, missing values are a challenge in many observational studies but are not generated by the simulator in its current form. Studying the effects of missingness on the ability of estimators to adjust for confounding bias is an interesting direction for future work. Finally, the estimators used to fit initial-state mechanisms were fit with a single set of hyperparameters. It is likely that the fit to Adult could be improved slightly by optimizing these.

Our benchmarking results show that models with comparable fit to observed variables can vary greatly in the quality of their causal effects estimates, highlighting the challenge of the tasks and the importance of developing purpose-built data sets, causal estimators, and model selection criteria.

## Acknowledgments and Disclosure of Funding

The authors would like to thank Anton Matsson for valuable discussions in the preparation of this manuscript and Sonja Johansson for providing the time and motivation to complete this work. We also acknowledge the students of the 2023 Nordic Probabilistic AI summer school who took part in a demo of the Income-SCM simulator. The work was supported by the Wallenberg AI, Autonomous Systems and Software Program (WASP), funded by the Knut & Alice Wallenberg Foundation.

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

# Appendix

## A  Additional information

1. URL to website/platform where the dataset/benchmark and metadata can be viewed and downloaded: `https://github.com/Healthy-AI/IncomeSCM/`

2. A full documentation of the data set's construction and intended is in our datasheet: `https://github.com/Healthy-AI/IncomeSCM/blob/main/datasheet.md`

3. The authors bear all responsibility in case the present work has violated rights or license terms for data used in this project.

4. The IncomeSCM project (both the simulator and data sets) will be hosted on Github. The data set will also be mirrored on `https://www.healthyai.se`. The raw data for the Adult data set is hosted on UCI Machine Learning Repository, as well as many mirror sites across the web. The authors will provide the necessary maintenance of the data set.

5. The data is stored as `.pkl` and `.csv` files of Pandas data frames (created by `df.to_pickle(...)` and `df.to_csv(...)`). The dataframes can be loaded into Python using `Pandas.read_pickle(...)` or `Pandas.read_csv(...)`

6. The license terms for IncomeSCM are described in Appendix B.1.

7. To ensure reproducibility, the authors aim to satisfy the ML Reproducibility Checklist: `https://arxiv.org/pdf/2003.12206`

8. Metadata in the Croissant format is included in `https://github.com/Healthy-AI/IncomeSCM/blob/main/metadata.json`

## B  Data set information

**Data sheet:** `https://github.com/Healthy-AI/IncomeSCM/blob/main/DATASHEET.md`

### B.1  License information

Both IncomeSCM data set samples and the simulator itself are shared under the Creative Commons Attribution 4.0 International (CC BY 4.0) license: `https://creativecommons.org/licenses/by/4.0/legalcode`. This allows for the sharing and adaptation of the datasets for any purpose, provided that the appropriate credit is given. The author of IncomeSCM is Fredrik D. Johansson.

The Adult data set by Barry Becker and Ronny Kohavi is licensed under a Creative Commons Attribution 4.0 International (CC BY 4.0) license. IncomeSCM does not host any part of the Adult data set and has made no permanent modifications to it.

## C  Data set details

Table 4: Baseline characteristics in the `IncomeSCM-1.0` cohort at $t = 0$ and the base data set Adult (after removing missing values), as well as in-sample fit quality. Categorical variables are reported as number (rate) and continuous variables as mean (interquartile range). The proportion of subjects with income higher than \$50 000 is higher in `IncomeSCM-1.0` than in Adult since the average income was calibrated to match the US population of 2023.

|  | IncomeSCM-1.0 ($n =$50000) | Adult ($n =$30162) |
|---|---|---|
| native-country |  |  |
| United-States | 45519 (91.0) | 27504 (91.2) |
| Philippines | 295 (0.6) | 188 (0.6) |
| England | 137 (0.3) | 86 (0.3) |
| Mexico | 1043 (2.1) | 610 (2.0) |
| Italy | 118 (0.2) | 68 (0.2) |
| China | 123 (0.2) | 68 (0.2) |

| | | |
|---|---|---|
| Vietnam | 86 (0.2) | 64 (0.2) |
| Dominican-Republic | 119 (0.2) | 67 (0.2) |
| India | 168 (0.3) | 100 (0.3) |
| Germany | 229 (0.5) | 128 (0.4) |
| Peru | 53 (0.1) | 30 (0.1) |
| France | 39 (0.1) | 27 (0.1) |
| Taiwan | 55 (0.1) | 42 (0.1) |
| El-Salvador | 185 (0.4) | 100 (0.3) |
| South | 109 (0.2) | 71 (0.2) |
| Puerto-Rico | 190 (0.4) | 109 (0.4) |
| Iran | 77 (0.2) | 42 (0.1) |
| Nicaragua | 56 (0.1) | 33 (0.1) |
| Honduras | 18 (0.0) | 12 (0.0) |
| Columbia | 108 (0.2) | 56 (0.2) |
| Canada | 189 (0.4) | 107 (0.4) |
| Portugal | 64 (0.1) | 34 (0.1) |
| Jamaica | 123 (0.2) | 80 (0.3) |
| Haiti | 71 (0.1) | 42 (0.1) |
| Scotland | 15 (0.0) | 11 (0.0) |
| Trinadad&Tobago | 33 (0.1) | 18 (0.1) |
| Outlying-US(Guam-USVI-etc) | 19 (0.0) | 14 (0.0) |
| Poland | 95 (0.2) | 56 (0.2) |
| Cambodia | 26 (0.1) | 18 (0.1) |
| Japan | 107 (0.2) | 59 (0.2) |
| Yugoslavia | 38 (0.1) | 16 (0.1) |
| Cuba | 137 (0.3) | 92 (0.3) |
| Hong | 26 (0.1) | 19 (0.1) |
| Ireland | 29 (0.1) | 24 (0.1) |
| Ecuador | 48 (0.1) | 27 (0.1) |
| Laos | 36 (0.1) | 17 (0.1) |
| Holand-Netherlands | 4 (0.0) | 1 (0.0) |
| Guatemala | 104 (0.2) | 63 (0.2) |
| Greece | 54 (0.1) | 29 (0.1) |
| Thailand | 29 (0.1) | 17 (0.1) |
| Hungary | 26 (0.1) | 13 (0.0) |
| sex | | |
| Female | 16206 (32.4) | 9782 (32.4) |
| Male | 33794 (67.6) | 20380 (67.6) |
| race | | |
| White | 42947 (85.9) | 25933 (86.0) |
| Black | 4677 (9.4) | 2817 (9.3) |
| Amer-Indian-Eskimo | 447 (0.9) | 286 (0.9) |
| Asian-Pac-Islander | 1529 (3.1) | 895 (3.0) |
| Other | 400 (0.8) | 231 (0.8) |
| age | 41.1 (32.0, 49.0) | 38.4 (28.0, 47.0) |
| education | | |
| Assoc-acdm | 1819 (3.6) | 1008 (3.3) |
| Bachelors | 8462 (16.9) | 5044 (16.7) |
| Masters | 2937 (5.9) | 1627 (5.4) |
| HS-grad | 15219 (30.4) | 9840 (32.6) |
| 10th | 1240 (2.5) | 820 (2.7) |
| Assoc-voc | 4503 (9.0) | 1307 (4.3) |
| Some-college | 9143 (18.3) | 6678 (22.1) |
| Prof-school | 1077 (2.2) | 542 (1.8) |
| 7th-8th | 881 (1.8) | 557 (1.8) |
| 9th | 785 (1.6) | 455 (1.5) |
| 5th-6th | 512 (1.0) | 288 (1.0) |
| 11th | 1525 (3.0) | 1048 (3.5) |
| 1st-4th | 272 (0.5) | 151 (0.5) |

| | | |
|---|---|---|
| Doctorate | 739 (1.5) | 375 (1.2) |
| 12th | 829 (1.7) | 377 (1.2) |
| Preschool | 57 (0.1) | 45 (0.1) |
| workclass | | |
| Private | 36639 (73.3) | 22286 (73.9) |
| Self-emp-inc | 1746 (3.5) | 1074 (3.6) |
| Local-gov | 3599 (7.2) | 2067 (6.9) |
| State-gov | 2183 (4.4) | 1279 (4.2) |
| Self-emp-not-inc | 4161 (8.3) | 2499 (8.3) |
| Federal-gov | 1661 (3.3) | 943 (3.1) |
| Without-pay | 11 (0.0) | 14 (0.0) |
| occupation | | |
| Tech-support | 1518 (3.0) | 912 (3.0) |
| Other-service | 5077 (10.2) | 3212 (10.6) |
| Exec-managerial | 6767 (13.5) | 3992 (13.2) |
| Machine-op-inspct | 3389 (6.8) | 1966 (6.5) |
| Transport-moving | 2685 (5.4) | 1572 (5.2) |
| Adm-clerical | 6164 (12.3) | 3721 (12.3) |
| Prof-specialty | 6955 (13.9) | 4038 (13.4) |
| Protective-serv | 1073 (2.1) | 644 (2.1) |
| Craft-repair | 6654 (13.3) | 4030 (13.4) |
| Handlers-cleaners | 2010 (4.0) | 1350 (4.5) |
| Sales | 5757 (11.5) | 3584 (11.9) |
| Farming-fishing | 1668 (3.3) | 989 (3.3) |
| Priv-house-serv | 265 (0.5) | 143 (0.5) |
| Armed-Forces | 18 (0.0) | 9 (0.0) |
| marital-status | | |
| Widowed | 1365 (2.7) | 827 (2.7) |
| Married | 26695 (53.4) | 14456 (47.9) |
| Never-married | 12766 (25.5) | 9726 (32.2) |
| Divorced | 7563 (15.1) | 4214 (14.0) |
| Separated | 1611 (3.2) | 939 (3.1) |
| relationship | | |
| Unmarried | 5201 (10.4) | 3212 (10.6) |
| Not-in-family | 13421 (26.8) | 7726 (25.6) |
| Husband | 17261 (34.5) | 12463 (41.3) |
| Wife | 6762 (13.5) | 1406 (4.7) |
| Other-relative | 1546 (3.1) | 889 (2.9) |
| Own-child | 5809 (11.6) | 4466 (14.8) |
| capital-net | 3766.3 (-8745.0, 16370.5) | 1003.6 (0.0, 0.0) |
| hours-per-week | 42.0 (26.0, 57.0) | 40.9 (40.0, 45.0) |
| education-num | 10.3 (9.0, 13.0) | 10.1 (9.0, 13.0) |
| income_prev | 55709.8 (13374.0, 76361.8) | – |
| studies_prev | | |
| Evening course | 16804 (33.6) | – |
| No studies | 24955 (49.9) | – |
| Full-time studies | 7509 (15.0) | – |
| Day course | 732 (1.5) | – |
| studies | | |
| No studies | 25934 (51.9) | – |
| Evening course | 18168 (36.3) | – |
| Full-time studies | 5275 (10.5) | – |
| Day course | 623 (1.2) | – |
| income | 93452.1 (45804.5, 126097.2) | – |
| income>50k | | |
| False | 30420 (60.8) | 22654 (75.1) |
| True | 19580 (39.2) | 7508 (24.9) |

Table 5: Commonly used benchmarks in causal effect estimation. Abbreviations: Arb. = arbitrary, Synth. = Synthetic, S-synth. = semi-synthetic, S-exp. = Semi-experimental., G.truth = Ground truth, C.graph = Causal graph, Unk. = Unknown, Rand. = Randomized, Des.=Designed, Data-d = Data-driven.

| Dataset | #Samples | #Cov | Context | Outcome | Treatment | G.truth | C.graph |
|---------|----------|------|---------|---------|-----------|---------|---------|
| IHDP | 747 | 25 | Real | Synth, | S-synth, | Yes | Unk. |
| Jobs | 3212 | 17 | Real | Real | S-exp | No | Unk. |
| Twins | 71344 | 46 | Real | Real | Real | No | Unk. |
| ACIC 2016 | 4802 | 58 | Real | Synth, | Synth, | Yes | Unk. |
| ACIC 2018 | 100 000 | 178 | Real | Synth, | Synth, | Yes | Rand. |
| IncomeSCM-v1 | Arb. (50k) | 11 | Data-d. | S-synth, | Synth, | Yes | Des. |

Table 6: Direct causes of each variable. A subscript of $_t$ indicates same-time causes and $_{t-1}$ previous-time causes.

| Abbr | Variable | Initial-state causes ($t = 0$) | Transition causes ($t > 0$) |
|------|----------|-------------------------------|------------------------------|
| age | Age of subject | – | $age_{t-1}$ |
| sex | Self-reported sex | – | $sex_{t-1}$ |
| rac | Self-reported race | – | $rac_{t-1}$ |
| nat | Native country | – | $nat_{t-1}$ |
| edu | Years/degree of education | age, rac, sex, nat | $edu_{t-1}, stu_{t-1}$ |
| wor | Workclass | age, edu, rac, sex, nat | $wor_{t-1}$ |
| mar | Marital status | age, edu, wor, rac, nat | $age_t, mar_{t-1}, stu_{t-1}$ |
| occ | Occupation | age, edu, wor, rac, sex, nat | $age_t, edu_t, wor_t, rac_t,$ $sex_t, nat_t, occ_{t-1}, stu_{t-1}$ |
| rel | Relationship | age, edu, wor, mar, rac, sex | $age_t, edu_t, wor_t, mar_t, rac_t,$ $sex_t, rel_{t-1}$ |
| hou | Work hours per week | age, edu, wor, mar, occ, rac, rel, sex | $age_t, edu_t, wor_t, mar_t r, occ_t,$ $rac_t, rel_t, sex_t, hou_{t-1}$ |
| cap | Capital net result | age, edu, wor, occ, mar, rac, rel, sex | $age_t, edu_t, wor_t, occ_t, mar_t,$ $rac_t, rel_t, sex_t, cap_{t-1}$ |
| stu | Studies | age, sex, edu, rel | $age_t, sex_t, edu_t, rel_t$ $inc_{t-1}, stu_{t-1}$ |
| inc | Yearly income | age, edu, wor, occ, mar, rac, sex, you, cap, stu | $age_t, edu_t, wor_t, occ_t, mar_t,$ $hou_t rac_t, sex_t, cap_t, stu_t,$ $inc_{t-1}, stu_{t-1}$ |

# D Commonly used benchmark data sets for causal effect estimation

See Table 5 for a summary of commonly used benchmarks for causal effect estimation.

# E Simulator details

- The edge list for initial-states and transitions are given in Table 6.
- The transition mechanisms for all variables are described in Table 7.

Table 7: Descriptions of handcrafted transition models. Some models (e.g., Education) have clipping operations to ensure that the variables stays in the specified range. Others have transitions that are too complex to write in this table. We refer the reader to the code for definition these.

| | |
|---|---|
| Age | Increase by one:
$\mathtt{age}_t = \mathtt{age}_{t-1} + 1$ |
| Sex | Constant:
$\mathtt{sex}_t = \mathtt{sex}_{t-1}$ |
| Race | Constant:
$\mathtt{rac}_t = \mathtt{rac}_{t-1}$ |
| Native country | Constant:
$\mathtt{nat}_t = \mathtt{nat}_{t-1}$ |
| Education | Increase education level with probability corresponding to studies
$p(\mathtt{edu}_t = \mathtt{edu}_{t+1} + 1) = 0.95\mathbb{1}[\mathtt{stu}_t = \mathtt{full}] + 0.05\mathbb{1}[\mathtt{stu}_t = \mathtt{evening}] + 0.1\mathbb{1}[\mathtt{stu}_t = \mathtt{day}]$ |
| Workclass | Stay with constant probability if not without pay or re-draw:
$s \sim \mathrm{Ber}(p_{\mathrm{stay}}); \; s' = s\mathbb{1}[\mathtt{wor}_{t-1} \neq \mathtt{without\_pay}]; \; \tilde{\mathtt{wor}}_t \sim \sigma(f_{\mathtt{wor}}(\mathbf{pa}(\mathtt{wor}_t)))$
$\mathtt{wor}_t = s'\mathtt{wor}_{t-1} + (1-s')\tilde{\mathtt{wor}}_t$ |
| Marital status | Change marital status according to possible transitions, age and studies.
E.g., "never married" to "divorced" is disallowed.
The probability of marrying is reduced for unmarried people engaged in full-time studies. |
| Occupation | High probability of staying in the same occupation except if finishing full-time studies
$p = p_{\mathrm{stay}}(\mathbb{1}[\mathtt{stu}_{t-1} \neq \mathtt{full}] + \frac{1}{4}\mathbb{1}[\mathtt{stu}_{t-1} = \mathtt{full}]); \; s \sim \mathrm{Bern}(p);$
$\tilde{\mathtt{occ}}_t \sim \sigma(f_{\mathtt{occ}}(\mathbf{pa}(\mathtt{occ}_t))); \; \mathtt{occ}_t = s \cdot \mathtt{occ}_{t-1} + (1-s)\tilde{\mathtt{occ}}_t$ |
| Relationship | Stay in relationship status with given probability $p_{\mathrm{stay}}$, sample new status from
initial-state model otherwise:
$s \sim \mathrm{Bern}(p_{\mathrm{stay}}); \; \tilde{\mathtt{mar}}_t \sim \sigma(f_{\mathtt{mar}}(\mathbf{pa}(\mathtt{mar}_t))); \; \mathtt{mar}_t = s \cdot \mathtt{mar}_{t-1} + (1-s)\tilde{\mathtt{mar}}_t$ |
| Hours-per-week | Perturbs the weekly hours by constructing a convex combination of the previous
hours with a draw from a the initial-state model of weekly hours given parents:
$\tilde{\mathtt{hou}}_t \sim \sigma(f_{\mathtt{hou}}(\mathbf{pa}(\mathtt{hou}_t))); \; \mathtt{hou}_t = \mathrm{clip}(\alpha \cdot \mathtt{hou}_{t-1} + (1-\alpha)\tilde{\mathtt{hou}}_t, 0, 7 \cdot 24)$ |
| Capital net | Samples whether there are capital gains/losses at all, depending on whether there was
last year. High probability of having some capital if there was capital last year.
If capital, samples a noisy version of last years value with high probability, otherwise a
new value from initial-state model. |
| Studies | High probability of staying within a program (e.g., Bachelor's) if started. Low probability
of starting full-time studies if not studying previously, or if already high income ($> 50\mathrm{k}$).
Zero probability of full-time studies if doctoral degree. |
| Income | Samples a new salary from initial-state model with current-time inputs if full-time studies
ended, if previously had no job, or if previous income was $< 5k$. Otherwise, add a random
raise to previous income depending on previous studies. Set income to 0 if current
full-time studies and to 4/5ths if engaged in day course. |

Table 8: Hyperparameter ranges for CATE estimators. Logistic regression, Ridge, Random Forest all used Scikit-learn implementations. XGBoost used the Python package "xgboost". Below are the corresponding hyperparameters used for estimators of causal effects. For all T-learners, the same ranges were used for both base estimators.

| Estimator | Hyperparameter | Range |
|---|---|---|
| IPW (LR) | C | [0.005, 0.01, 0.05, 0.1, 0.5, 1, 5, 10, 50, 100] |
| IPWW (LR) | C | [0.005, 0.01, 0.05, 0.1, 0.5, 1, 5, 10, 50, 100] |
| IPW (RF) | min_samples_leaf | [5, 10, 20, 50, 100] |
| IPWW (RF) | min_samples_leaf | [5, 10, 20, 50, 100] |
| S-learner (Ridge) | C | [0.01, 0.02, 0.1, 0.2, 1, 2, 10, 20, 100, 200, 1000] |
| T-learner (Ridge) | C | [0.01, 0.02, 0.1, 0.2, 1, 2, 10, 20, 100, 200, 1000] |
| S-learner (RF) | min_samples_leaf | [5, 10, 20, 50, 100] |
| T-learner (RF) | min_samples_leaf | [5, 10, 20, 50, 100] |
| S-learner (XGB) | tree_method | ['hist'] |
|  | eta | [0.1, 0.3, 0.5, 0.7] |
|  | max_depth | [3, 5, 7, 9] |
| T-learner (XGB) | tree_method | ['hist'] |
|  | eta | [0.1, 0.3, 0.5, 0.7] |
|  | max_depth | [3, 5, 7, 9] |

## F   Estimator details

The hyperparameter ranges for all estimators are given in Table 8. For all regression and matching estimators and nuisance regressions, hyperparameters were selected based on the $R^2$ score, evaluated in 5-fold cross-validation. For all nuisance classifiers (e.g., propensity scores estimators), hyperparameters were selected based on the AUROC in 5-fold cross-validation.

Regression adjustment estimators directly model the outcome $Y$ as a function of the intervention $A$ and adjustment variables $Z$ which guarantee identifiability. Among these, we include T-learners, which fit one model per *potential outcome* (intervention), and S-learners, which treat the intervention as an input to a single model, mixed in with adjustment variables (Künzel et al., 2019). T-learners use a different base estimator $\hat{m u}_a(z)$ to model the potential outcome $Y(a)$ of each intervention $a$, as a function of a chosen adjustment set $Z$. The CATE is then estimated as

$$\hat{\text{CATE}}(z) = \hat{\mu}_1(z) - \hat{\mu}_0(z).$$

S-learners fit a single base estimator $\hat{\mu}(z, a)$ as a function of both the adjustment set and the intervention. They estimate the CATE as

$$\hat{\text{CATE}}(z) = \hat{\mu}(z, 1) - \hat{\mu}(z, 0).$$

and both estimate the ATE for a set of $n$ samples with adjustment set covariates $z_i$ as

$$\hat{\text{ATE}} = \frac{1}{n} \sum_{i=1}^{n} \hat{\text{CATE}}(z_i)$$

Inverse propensity-weighting (IPW) estimators model the intervention variable instead, as a function of the adjustment set, and re-weigh outcomes to estimate treatment effects. We use two classical estimators, the Horvitz-Thompson (IPW) and Hayek or "Weighted" (IPW-W) estimators of the average treatment effect (ATE). Both fit a function $\hat{e}(z) \approx p(A = 1 \mid Z = z)$ which is used to re-weight observed outcomes to form an estimate of the potential outcomes and the causal effect. Let $a_i \in \{0, 1\}$ indicate the binary intervention for subject $i$, $z_i$ their covariates from the adjustment set and $y_i$ the observed outcome corresponding to $a_i$.

$$\hat{\mu}_0 = \frac{1}{n} \sum_{i=1}^{n} w_i^0 y_i \tag{1}$$

$$\hat{\mu}_1 = \frac{1}{n} \sum_{i=1}^{n} w_i^1 y_i \tag{2}$$

where

$$w_i^0 = \frac{1 - a_i}{1 - e(z_i)} \quad w_i^1 = \frac{a_i}{e(z_i)}$$

for the Horvitz-Thompson estimator and

$$w_i^0 = \frac{1 - a_j}{1 - e(z_j)} \Big/ \left( \sum_j \frac{1 - a_j}{1 - e(z_j)} \right) \quad w_i^1 = \frac{a_i}{e(z_i)} \Big/ \left( \sum_j \frac{a_j}{e(z_j)} \right).$$

for the Hayek estimator. The ATE is then estimated as

$$\hat{\text{ATE}} = \hat{\mu}_1 - \hat{\mu}_0.$$

Finally, we include a nearest-neighbor matching estimator with a Euclidean metric. The estimator computes the nearest neighbor $j(i)$ for all samples $i \in [n]$ according to the chosen metric and estimates,

$$\hat{\mu}_0 = \frac{1}{n} \sum_{i=1}^{n} \left[ (1 - a_i)y_i + a_i y_{j(i)} \right] \tag{3}$$

$$\hat{\mu}_1 = \frac{1}{n} \sum_{i=1}^{n} \left[ a_i y_i + (1 - a_i)y_{j(i)} \right] \tag{4}$$

and

$$\hat{\text{ATE}} = \hat{\mu}_1 - \hat{\mu}_0.$$

For Tasks 1–2, all estimators use the same conditioning set as the adjustment set. For Task 3, the CATE is estimated as in Task 1 but stratified and aggregated in 16 numeric bins of `education`. Ground truth for each task is constructed analogously but with sampled potential outcomes, generated by the simulator for each observation, in place of $\hat{\mu}$.

