# OpenReview forum: "IncomeSCM: From tabular data set to time-series simulator and causal estimation benchmark"
_NeurIPS.cc/2024/Datasets_and_Benchmarks_Track — NeurIPS 2024 Track Datasets and Benchmarks Poster_

### Official Review · Reviewer_dKd7 · 2024-07-18
**Feedback**

**Rating:** 7
**Confidence:** 4
**Correctness:** Some assumptions need justification. …
**Clarity:** The paper is well-written and easy to…

**Review:**

**Pros:**

This paper addresses an important problem in the machine learning community, particularly in causal inference. The lack of sufficient benchmarks with interventional ground truth has made the evaluation of developing algorithms challenging and sometimes uninformative.

The paper comprehensively covers the limitations of current benchmarks, both simulator-based and real-world. The former often lacks real-world nuance, while the latter suffers from an inability to perform manipulations or small sample sizes when such manipulations are performed through techniques like subsampling.


**Cons:**

The assumption that the transition mechanisms are significantly simpler than the mechanisms among observational variables needs justification, either theoretically or with numerous concrete examples from real-world systems. Can you mention any real-world counterexamples of this assumption in the real world? It would also be beneficial to explain under which circumstances the time-homogeneity of the transition causal mechanisms is a fair assumption.
Why are the constructed variables fully dependent on other variables at the initial time? It seems they are excluded from the input of the $h_R$ function, meaning that they do not provide additional information for income. However, in the transition mechanism, these variables affect the income variable at the same time. Why does this reasoning not apply here?

**Strengths:**

Tackling a critical problem in the community of causal inference.

**Additional Feedback:**

No.

**Documentation:**

Yes.

**Ethics:**

No.

**Limitations:**

I mentioned above.

**Opportunities For Improvement:**

Provide justifications for the assumptions as requested in the feedback.

**Relation To Prior Work:**

The limitations of ther related work are well discussed.

**Summary And Contributions:**

This work aims to design a generalizable method to create benchmarks from observational data to evaluate causal estimation methods. The complexity of the benchmark arises from the sequential construction of the causal graph, where each transition step is simple, but the accumulation of these steps results in a complex functional relationship from an early cause to a downstream effect in the temporal sequence. The simulator fits the observational data at initial time and then generates data for later time points through hand-designed transition mechanisms.

---

> ### Author Rebuttal · Authors · 2024-08-14
>
> Thank you for a very thoughtful review and many good suggestions.
>
> **R: The assumption that the transition mechanisms are significantly simpler than the mechanisms among observational variables needs justification, either theoretically or with numerous concrete examples from real-world systems. Can you mention any real-world counterexamples of this assumption in the real world? It would also be beneficial to explain under which circumstances the time-homogeneity of the transition causal mechanisms is a fair assumption. Why are the constructed variables fully dependent on other variables at the initial time? It seems they are excluded from the input of the function, meaning that they do not provide additional information for income. However, in the transition mechanism, these variables affect the income variable at the same time. Why does this reasoning not apply here?**
>
> * The strongest argument for this assumption is that most variables are easier to model once you know a recent value of them. The classical example is that the weather tomorrow is usually similar to the weather today (transition is simple), but modeling the weather based on other variables such as location and time-of-year is more difficult (initial-state model is complex). Another example is the distribution of patients in a hospital. When a new patient gets admitted (initial state), we can view their demographics, symptoms, and medications as drawn from a complex distribution of possible patients. When we observe them the next day, the changes in their variable tend to be more predictable given their values at admission and obey simpler rules, such as dose-response to medication or day-night cycles of vitals. A general way to think about this is that the dependence of a variable on other variables weakens once we know its previous value.
>
> * Prompted by your question, we will modify the discussion of the simplicity of transitions to clarify that certain changes are easy to predict (with high probability, you stay in your job) but other changes are difficult (if you change your job, what will be your next one?). In the simulator, we reuse the initial-state models within transition mechanisms for exactly this purpose, as discussed on p.5, right before Section 4.
>
> * A counter-example of this is bandit-like processes (e.g., slot machines) where the variables at time $t$ are independent of the variables at time $t-1$. The payout of the slot machine follows the same distribution at all time points.
>
> * I am not sure how to interpret the last question, perhaps you could expand? The only constructed variable is {\tt studies} and the value of this variable at all times $t$ affects (provides information for) the income at $t$. In fact, nearly all variables at a time $t$ affect the income at that time.

---

> > ### Comment · Reviewer_dKd7 · 2024-08-26
> > **Follow-up Comments**
> >
> > Thanks to the authors for their detailed responses to my questions. I have a few more follow-up comments that might help clarify things further.
> >
> > > A counter-example of this is bandit-like processes (e.g., slot machines) where the variables at time
> >  are independent of the variables at time
> > . The payout of the slot machine follows the same distribution at all time points.
> >
> > When I mentioned a "counter-example," I was thinking of cases where the transition dynamics are more complex than the overall flow (the sum of all transitions). It’s possible to have mathematical models where transitions cancel each other out, leading to a simpler overall path—like a random walk that ends up back at its starting point. Even though such systems might be rare in real life (happening with measure zero), it would be helpful to see the theoretical reasoning behind this assumption in addition to the intuition you’ve provided.
> >
> > > I am not sure how to interpret the last question, perhaps you could expand? The only constructed variable is {\tt studies} and the value of this variable at all times
> >  affects (provides information for) the income at
> > . In fact, nearly all variables at a time
> >  affect the income at that time.
> >
> > In the second paragraph of Section 3.1, it says,
> >
> > "For IncomeSCM, we construct a continuous income variable $Y$ by first fitting a small random forest regressor $h_R$ to predict the original binary outcome $R$ as a function of all covariates $X$ (excluding the added variable studies)."
> >
> > But in Table 5 of the Appendix, under "Simulator Details," the variable stu is listed as a cause of inc at the initial state, and stu_t is listed as a cause for inc_t. If I’m understanding this right, the text suggests that stu is excluded as a cause of the income variable, but the table shows it as a cause. This seems inconsistent to me, and I’d appreciate some clarification.
> >
> > In general, I appreciate the concept of combining observational data with hand-designed models to create benchmarks with varying levels of complexity, which is reflected in my initial high score. I maintain my score and hope that these comments help further improve the paper.

---

> > > ### Author Response · Authors · 2024-08-26
> > > **Re:**
> > >
> > > Thank you for engaging with the rebuttal and for maintaining your score!
> > >
> > > 1. I see what you mean by counter-example now—although the assumption is only about the initial state, not the *overall flow*. A general counter-example would be one where all of the variables are independently set in the initial state but become dependent in transitions. For example, let the variables measure the opinions of a random set of people (e.g., students) who begin to meet regularly (say in a class). Since the initial set is random, there will be no correlation among the opinions and the initial-state model is fully random. The next state may be a complicated function of the opinions of different people.
> > >
> > > 2. Thank you for clarifying. The income variable is computed by first estimating the income of an individual that is not studying (random forest regressor fit to the other parent variables) and that income is then modified based on the study pace. For simplicity: full income if no studies, 0% if full-time studies, 80% if taking a day course once a week. We will clarify this in the paper.

---

### Official Review · Reviewer_rn5w · 2024-07-19
**Reasonable approach, but difficult to generalize**

**Rating:** 6
**Confidence:** 4
**Clarity:** It is very clearly written, but lacks…

**Review:**

The paper is mostly clearly written, but lacks a few more details on, e.g., how the hand-crafted models look like exactly. That being said, it remained unclear whether a time-series dataset is created or if the time dynamics are just used to simulate new outcomes after T years, i.e., the generated data remains IID, just the generation process utilizes dynamics. Generally, the proposed methodology is general enough to be applied to different real-world datasets, although not necessarily novel and requires some significant domain knowledge. The procedure (aside from the hand-crafted models) is sufficiently explained, and the GitHub page is nicely documented. Regarding the experiments, the insight that different estimators perform differently is less helpful; perhaps the phrasing could be more towards emphasizing that the generated datasets are not trivial. The main concern here is that the requirement of hand-crafted models would be a strong requirement for generalization to other datasets.

See below for other points and questions.

**Strengths:**

- The method is general enough for other datasets (although requires significant hand-crafting).
- There is a very good related work comparison and discussion.
- The documentation on GitHub is great.
- The writing is clear, although some details are missing.

**Additional Feedback:**

Generally, the idea makes sense, although not necessarily novel. This may not be a strong requirement here, but my main concern is the current requirement of hand-crafted models and knowledge about the graph. I see the issue with generating 'arbitrary' relationships (e.g., divergence after a few steps, collapsing, etc.), but requiring domain knowledge to apply the method to new datasets is a significant issue in practice. Without easy practical generalization, only having one dataset is too limited. Maybe the author can briefly comment on this, as I might be missing some details that would simplify this. Other remarks in addition to the ones above:

- For the graph representation, consider using an 'unrolled' representation like in Figure 10.1(+) in the book "Elements of Causal Inference".
- It is unclear why you have f_v != g_v, what is the reason behind not having a consistent mechanism here?
- The general idea of representing this as a dynamic process to generate new data is not novel, but rather the concrete process for the Adult dataset. Here, the paper could be strengthened by adding more datasets, especially since it requires some hand-crafted modeling.
- It is unclear whether the regression on the binary outcome makes sense. Why is this not simply kept as binary? Since it indicates whether the income is above or below $50,000, what would a value of, e.g., "0.7" mean?
- The way how categorical causal mechanisms are modeled could be improved. For instance, by just fitting a model that provides the conditional probabilities, one can simply sample based on the conditional probabilities of classes given a specific parent observation. This way, the 'noise' here is the probability of having a specific class given a parent sample. This might be closer to the additive noise model equivalent without restricting it to classification models that require some continuous representation.
- It could be insightful to have a time series plot to see how the dynamic develops exemplary.

**Correctness:**

The proposed approach is correct and makes sense from a causal perspective, although the categorical modeling could be improved (see below).

**Documentation:**

The GitHub page has good documentation, but the paper lacks details about the hand-crafted models.

**Ethics:**

Slight concern regarding the use of the Adult dataset. However, this is an old and well-established dataset. Therefore, the concern is rather minor.

**Limitations:**

The author discusses some of the limitations, but it lacks more discussion on choosing hand-crafted models over generating them. This is a bottleneck to effectively generalizing it to other datasets.

**Opportunities For Improvement:**

- The introduction and related work sections have a large overlap. Perhaps the related work introduced in Section 1 could be merged with Section 2.
- There is a lack of details on how the handcrafted models look like. In particular, there is not enough discussion or motivation as to why one would not try to generate these models (e.g., using a multilayer perceptron with random weights). The author could perhaps use arguments like the divergence of values without proper control, etc.
- Due to the large focus on time dynamics, it is unclear whether the resulting dataset is still independent and identically distributed (IID) (i.e., the time logic is applied to generate a single row) or if the dataset becomes a time series. This could be pointed out more clearly.
- While evaluating treatment effect results makes sense as an application of the generated data, the focus here could be more on showing that the data is non-trivial rather than demonstrating that different methods perform differently. In this regard, commonly used DML methods are missing.

**Relation To Prior Work:**

Great comparison to related work.

**Summary And Contributions:**

The author describes a general approach to generate new synthetic datasets based on real-world data by explicitly creating an SCM modeling time dependencies. Here, the causal mechanisms are a mix of inferred models and various hand-crafted models. This has been applied to the real-world Adult dataset that contains data about incomes given certain features. The generated datasets are used to compare different causal estimators.

---

> ### Author Rebuttal · Authors · 2024-08-14
>
> Thank you for a very thoughtful review and many good suggestions.
>
> **R: Lack of details on the handcrafted models. [...] Why not try to generate them?**
>
> *  We describe some handcrafted models at the end of p.5, right before Section 4. For example, the relationship variable remains unchanged with some probability and is drawn anew from the initial-state model ( with inputs from the current time). Income transitions depend on whether a subject was previously working or studying, what their previous salary was, and a random raise. We have constructed a table describing all variables (see attached .pdf) to include in the appendix.
>
> * Generating random mechanisms is an interesting suggestion, and was explored in the ACIC 2018 competition [1]. Their biggest drawback is low realism (e.g., divergence). Our dynamics are designed to capture real patterns (age goes up by 1, salary tends to increase), and stay close to the observed statistics (using initial-state models when relationship changes). An interesting direction is to make generated mechanisms obey handcrafted constraints!
>
> **R: My main concern is the current requirement of hand-crafted models and knowledge about the graph.**
>
> Yes, a new application would require a new causal graph but we argue that it is worth the effort! Having a graph is a strong *benefit*: Causal inference fundamentally requires assumptions but most benchmarks cannot validate them. Many estimators use adjustment sets (regression adjustment, propensity modeling, DML) to control for confounders. For these, it is important to know **which** adjustment sets are valid for the identification of causal effects. A causal graph can validate sets through graphical criteria (backdoor criterion, IV, proxies). Further,  since the graph must be postulated, users of the simulator can inspect and criticize the assumptions made by its creator. Generating random graphs will make this more difficult.
>
> **R: While evaluating treatment effect results makes sense [...], the focus could be more on showing that the data is non-trivial rather than demonstrating that different methods perform differently. [...] DML methods are missing.**
>
> * This is a good challenge. Defining 'non-trivial' is difficult. For CATE estimation, it should be tied to the difficulty of ranking subjects in terms of their causal effect. We provide empirical evidence for the non-triviality of this task in Table 2 through the low CATE $R^2$ values achieved by all models, despite a fairly large sample size. This is further echoed in the stratification of effects by education in Table 3, where most methods are off by tens of thousands of dollars.
>
> * Below, we give the results of DML CATE estimation created by fitting regressions to the target $\frac{y_i - M_Y(x_i)}{a_i - M_A(x_i)}$ with sample weights $w_i = (a_i - M_A(x_i))^2$ where $M_Y(x)$ models the $Y$ and $M_A(x)$ models actions $A$ as functions of $X$. This stems from the residualization of [2], as used by [3].
>
> | Estimator | $R^2$ CATE ($\uparrow$) | AE ATE ($\downarrow$) | AUC/$R^2$ CV ($\uparrow$) |
> | --- | --- | --- | --- |
> | DML (Linear) | -0.02 (-0.02, -0.02) | 8299 (7784, 8817) | 0.79 |
> | DML (XGB) | -0.25 (-0.26, -0.23) | 4580 (4012, 5199) | 0.83 |
> | DML (Mix) | 0.06 (0.06, 0.07) | 6069 (5575, 6581) | 0.83 |
>
> DML (Mix) uses an XGB regressor for $M_Y$, logistic regression for $M_T$ and ridge regression for the CATE model. The poor results are likely due to high variance induced by weighting.
>
> **R: [...] It is unclear whether the resulting dataset is still IID.**
>
> Each row of the data set represents a single subject's sequence,  which is IID to the sequences (rows) of other subjects (the reviewer's first option). This will be clarified.
>
> **R: It is unclear why $f_v \neq g_v$**
>
> Foremost, this is because the parents of a variable $v$ differ between its initial state ($f_v$-9 and its transitions )$g_v$). Most variables $v_t$ depend on the previous state $v_{t-1}$. In the initial state, there is no previous state and $f_v$ cannot have $v_{t-1}$ as input. The mechanisms must behave quite differently; $g_v$ describes the change from the previous value, and $f_v$ the initial value. We will clarify this.
>
> **R: It is unclear whether the regression on the binary outcome makes sense. [...] What would a value of, e.g., "0.7" mean?**
>
> * Leaving the outcome binary would make it more difficult to design, for example, income transitions. With a continuous value, we can reason about the average salary for jobs and the increase due to a pay raise. A binary-income model would be much more coarse-grained.
>
> * A value of $0.7$ indicates how likely the income is to exceed \$50 000. For subjects where the model outputs a higher number, the continuous income is higher. This is rescaled and shifted so that the resulting values roughly match the income distribution in the US population, see caption of Table 2. We have expanded this discussion in the final version.
>
> **R: The way categorical causal mechanisms are modeled could be improved. [...] one can simply sample based on the conditional probabilities of classes given a specific parent observation.**
>
> This is exactly what we do: categorical variables are sampled from the conditional probabilities given by probabilistic classifiers such as logistic regression. The Gumbel-Max procedure is equivalent, as we mention on p.5. We used this description to write both mechanism types on structural equation (SEM) form, see p.27 in [4]. We have corrected a typo that said that $f(...)$ returns the probabilities instead of the logits.
>
> We thank Reviewer rn5w for all suggestions and plan to add the new version of Adult to a future release!
>
>
> [1] Shimoni, et al. Benchmarking framework for performance-evaluation of causal inference analysis.  2018
>
> [2] Robinson. Root-N-consistent semiparametric regression. 1988
>
> [3] Chernozhukov, et al. Double/debiased machine learning for treatment and causal parameters. 2017
>
> [4] Pearl. Causality. 2009

---

> > ### Comment · Reviewer_rn5w · 2024-08-14
> >
> > I want to thank the authors for their thorough response. They address all my technical concerns. My remaining concern, however, is that it only introduces a single new type of dataset that is semi-synthetic (although with varying parameters). Here, a single dataset is fine if it is collected from a real-world process, but in the case of semi-synthetic data, I would expect the application of the method to multiple different datasets. Regarding the point "requirement of hand-crafted models and knowledge about the graph," the concern is less about knowledge of the graph since this is a common requirement in causal tasks, but more about the hand-crafted causal mechanisms to generate a new dataset. This makes it practically unlikely that someone would build on top of this framework to apply it to a new dataset. However, because this is more of a subjective opinion, I will still raise my score to 6, but do not feel comfortable going higher. Once again, I want to thank the authors again for taking the time to address each of my points.

---

> > > ### Author Response · Authors · 2024-08-15
> > > **Re:**
> > >
> > > Thank you for engaging with the rebuttal and for acknowledging our improvements!
> > >
> > > It is fair to raise concerns about reusability for this kind of work. We have done our utmost to make the code reusable by putting all task-specific (hand-crafted) components in a single Python file (income/income_samplers.py) and limiting these to a single function per variable (the folder "income" is general but poorly named, it will be renamed to "sim" or similar). The simulator configuration file (configs/simulator.yml) specifies which handcrafted function should be called for which variable and which parent variables should be made available in the data frame passed by the task-agnostic part of the code. Thus, for similar tasks (but with different data sets), only configuration files and the single sampler script need to be changed.

---

### Official Review · Reviewer_6W4j · 2024-07-23
**Interesting idea for benchmarking causal effect estimation but lacking further details and use-cases.**

**Rating:** 5
**Confidence:** 4

**Review:**

The paper presents a very interesting idea for generating semi-synthetic data for benchmarking causal effects. The idea is simple and thus easy to generalize, while producing more realistic datasets than those often used in the literature. However, the paper could significantly benefit from i) more details on the resulting dataset so that it is more clear how to use it; ii) exploring either datasets or variations of the interventions so that it is clear that the proposed simulator can be used beyond the setting shown in the paper; and iii) empirical comparison with the results on other datasets used in the literature to show more clear case on the need of the proposed benchmark.

**Strengths:**

- The simulator using a Markov process is very intuitive and thus allows an easy combination of data and human design for semi-synthetic datasets for causal effect estimation.
- The introduction and related work seem relevant and reasonably complete.
- The details of the simulator construction are clear and very intuitive.

**Additional Feedback:**

Maybe the the author should consider moving away from the original adult data to its new version: https://openreview.net/forum?id=bYi_2708mKK

Such versions cleans some of the non-informative/corrupted features and describe other tasks beyong income prediction that could be of interest for thge author to extend the simulator.

**Clarity:**

Overall the paper is well written but some further details could significantly improve its applicability and impact (see "Opportunities For Improvement"). More concrete examples are the paragraph above Section 5, where it remains unclear to me what exact data and how data are use un the different described empirical settings. Similarly, the sentence introducing CATE in page 4 would benefit from some rewriting.

**Correctness:**

I have not detected any correctness issues but only some typos here an there. The most important one is that in page 6, the second mention to "seq_parents_curr" should instead be "seq_parents_prev" if I am not mistaken.

**Documentation:**

The documentation looks good to me but could benefit from additional demos (currently missing in the repository) to show people how to use the simulator beyond the current benchmark.

**Ethics:**

No.

**Limitations:**

The author clearly states the limitations of the proposed simulator with regards to time horizon, censoring, and missing data. I believe these are all relevant problems  to the application context of the benchmark and thus I very much appreciated the discussion and hope to see future work on that direction.

As a side note based on my curiosity, regarding the former,   it is not clear to me what implication from a causal perspective would have to have heterogenous time horizons in the dataset. In such a case should the snapshot of the resulting dataset contain a proxy for time so that causal inference is doable?

**Opportunities For Improvement:**

- Further details on the resulting cross-sectional benchmark dataset, describing which variables are affected by hidden confounders and a result of the temporal feedback loop in the simulator, as well as a more careful description of how to properly use it for causal effect estimation.
- Intuitive and empirical comparison with other datasets previously proposed in the literature to solve these tasks. It remains partially unclear what this benchmark offers compared to those described in the related work. If it is the sample size of the data and its effect on the quality of the estimates and thus the comparison between methods, please show. If it is the generalizability of the proposed simulator, then show some additional examples.
- As a side comment, I wonder if the author should consider moving away from the original adult data to its new version: https://openreview.net/forum?id=bYi_2708mKK

**Relation To Prior Work:**

The related work section reads good to me.

**Summary And Contributions:**

The paper proposes a simulator to generate semisynthetic data for benchmarking causal effect estimators based on the Adult dataset. To this end, the author considers a Markov process that is initialized at time t=0 with the observation distribution learned from the (extended) Adult dataset to then simulate an intervention at a later time, which is finally used at a fixed time t=T to estimate the (cross-sectional) treatment effect. The proposed semi-synthetic data generation process aims to use both real-world data and expert knowledge (handcrafted in additional endogenous variables as well as in time dynamics) to generate more realistic benchmarks for causal inference. The resulting benchmark dataset is then used to estimate both the conditional average treatment effect (CATE) and the average treatment effect (ATE) under different assumptions about the data used for estimation.

---

> ### Author Rebuttal · Authors · 2024-08-14
>
> Thank you for a very thoughtful review and many good suggestions!
>
> **R: Further details on the resulting cross-sectional benchmark dataset, describing which variables are affected by hidden confounders and a result of the temporal feedback loop in the simulator, as well as a more careful description of how to properly use it for causal effect estimation.**
>
> * The resulting cross-sectional dataset has no unobserved confounders by default. More precisely, all direct causes of the treatment selection are included in the data set. As a result, there exist several sufficient adjustment sets for all tasks included in the IncomeSCM-v1 benchmarks, as discussed at the end of page 7, just before Section 5. This will be clarified further. For Task 3, the conditioning set cannot be used alone for adjustment since confounders are left out of this set.
>
> * I am not completely sure what you mean by a temporal feedback loop in this case. Could you please clarify? The sequential nature of the simulator does *not* introduce confounding for the benchmark tasks, as described above.
>
> * The description of how to use the data set for causal effect estimation (currently on page 7) will be expanded for the final version. We base our description on 'adjustment sets' since such sets are sufficient to implement propensity-based estimators, regression adjustment and doubly robust/double machine learning methods.
>
> **R: Intuitive and empirical comparison with other datasets previously proposed in the literature to solve these tasks. It remains partially unclear what this benchmark offers compared to those described in the related work. If it is the sample size of the data and its effect on the quality of the estimates and thus the comparison between methods, please show. If it is the generalizability of the proposed simulator, then show some additional examples.**
>
> * The main benefit of this simulator is that it offers a) higher realism than synthetic benchmarks that use simple mathematical functions (IHDP) or randomized functions (e.g., ACIC 2017, DoWhy) to generate outcomes, b) greater control over causal mechanisms than purely data-driven simulators (e.g., MedKit-Learn) by using a causal graph and per-variable mechanisms, and c) means to validate identification assumptions are valid, unlike non-synthetic benchmarks (e.g., Twins). We discuss this in Section 2 and contrast our approach to the literature in the Contributions section in Section 1. We will clarify this distinction further in the discussion.
>
> * Ranking data sets based on empirical metrics is difficult since different tasks are complementary and present different challenges. If we pick a metric for comparison (e.g., sample size), it is very easy to optimize this particular feature without yielding an interesting simulator. We have focused on combining realism (data-driven context variables) and control (hand-crafted dynamics and interventional distributions) which are both important aspects of causal estimation tasks. Being able to generate arbitrary sample sizes is also beneficial. Below is a summary of commonly used causal estimation data sets for inclusion in the final version of the paper.
>
> | Dataset | \#Samples | \#Covariates | Context | Outcome | Treatment | Ground truth | Causal graph |
> | --- | --- | --- | --- | --- | --- | --- | --- |
> | IHDP | 747 | 25 | Real | Synthetic | Semi-synthetic | Yes | Unknown |
> | Jobs | 3212 | 17 | Real | Real | Semi-experimental | No | Unknown |
> | Twins | 71344 | 46 | Real | Real | Real | No | Unknown |
> | ACIC 2016 | 4802 | 58 | Real | Synthetic | Synthetic | Yes | Unknown |
> | ACIC 2018 | 100 000 | 178 | Real | Synthetic | Synthetic | Yes | Randomized |
> | IncomeSCM-v1 | Arbitrary (50 000) | 11 | Data-driven | Semi-synthetic | Synthetic | Yes | Designed |
>
> **R: As a side note based on my curiosity, [...], it is not clear to me what implication from a causal perspective would have to have heterogenous time horizons in the dataset. In such a case should the snapshot of the resulting dataset contain a proxy for time so that causal inference is doable?**
>
> * Interesting question! First we would need to determine the causal parameter of interest. A reasonable choice is to think of the time horizon $T$ as an effect moderator and aim to identify $\mathbb{E}[Y(a) \mid T=t, X=x]$, that is, the expected income level after $t$ years for a subject with baseline variables $x$ that was intervened on with $A_1\leftarrow a$ at time $1$. Then, the data set could contain the time at which the outcome was recorded, in addition to the outcome itself. Depending on how horizons were determined in the data sets (which outcomes were recorded), causal identifiability may or may not be preserved without change. If the horizon was determined by an unobserved variable that also influenced the treatment assignment, this variable could be an unobserved confounder. If the horizon was random, identifiability would hold as-is.
>
> **R: As a side comment, I wonder if the author should consider moving away from the original adult data to its new version.**
>
> * This is a great suggestion, we will add this to the next version of the benchmark!
>
> We thank the reviewer for suggesting further improvements to the manuscript with regards to language and typos. These have been corrected for the final version.

---

### Author Rebuttal · Authors · 2024-08-14

We would like to thank all reviewers and chairs for evaluating our work. The reviews were thoroughly inspiring and we are confident that our rebuttal responds well to the brought-up concerns. We have addressed the opportunities for improvement suggested by reviewers, here in the rebuttals and in the final version of the paper:

* Details on the observational data set (Rev 6W4j). Improved description and clarification regarding (lack of) temporal confounding.
* Intuitive comparison with other data sets (Rev 6w4j). Added table comparing common benchmarks.
* Overlap between introduction and related work (Rev rn5w). Merged parts of related work from intro to related work and refer to table of common benchmarks.
* Lack of details on handcrafted models (Rev rn5w). Added table describing each transition mechanism (see rebuttal).
* Clarity regarding IID (Rev rn5w). Clarified in rebuttal and paper.
* Missing DML methods (Rev rn5w). Added results for common linear and non-parametric DML methods (see rebuttal).
* Justification of assumptions (Rev dkd7). Gave more examples of when transitions are simple, relative to initial-state models, and when they are not. Added discussion related to reuse of initial-state models in transitions.

We hope that these improvements and our individual rebuttals below will be well-received.

---

### Decision · Program_Chairs · 2024-09-26

**Decision:**

Accept (Poster)

**Comment:**

The paper proposes a novel simulator to generate semi-synthetic data for benchmarking causal effect estimators based on the Adult dataset. The approach starts with a Markov process that is initialized at time t=0 with the observation distribution learned from the (extended) Adult dataset. It then simulates an intervention at a later time, and uses it at a later fixed time t=T to estimate the (cross-sectional) treatment effect. The proposed semi-synthetic data generation process aims to use both real-world data and expert knowledge (handcrafted in additional endogenous variables as well as in time dynamics) to generate more realistic benchmarks for causal inference. The resulting benchmark dataset is then used to estimate both the conditional average treatment effect (CATE) and the average treatment effect (ATE) under different assumptions about the data used for estimation. The paper is very clear and the framework presentation style is rigorous.

Due to the simplicity of the approach, it can be generalized in a fairly straightforward manner. Furthermore, the paper generates more realistic datasets than those often used in the literature. Suggestions for paper improvement include: explore more datasets and/or more interventions, to more extensively demonstrate that the proposed simulator can be used beyond the setting shown in the paper.

The author's responses and the reviewers comments and the follow up responses were constructive and engaged, a pleasure to read!